# Ultra-Wideband Electromagnetic Composite Absorber Based on Pixelated Metasurface with Optimization Algorithm

**DOI:** 10.3390/ma16175916

**Published:** 2023-08-29

**Authors:** Changhyeong Lee, Kichul Kim, Pyoungwon Park, Yunseok Jang, Jeongdai Jo, Taein Choi, Hakjoo Lee

**Affiliations:** 1Research and Development Team, Center for Advanced Metamaterials, Daejeon 34103, Republic of Korea; 2Department of Flexible & Printed Electronics, Korea Institute of Machinery and Materials, Daejeon 34103, Republic of Korea

**Keywords:** absorber, composite, genetic algorithm, metamaterials, metasurfaces, ultra-wideband

## Abstract

An ultra-wideband electromagnetic (EM) absorber is proposed. The proposed absorber consists of two thin metasurfaces, four dielectric layers, a glass fiber reinforced polymer (GFRP), and a carbon fiber reinforced polymer (CFRP) which works as a conductive reflector. The thin metasurfaces are accomplished with 1-bit pixelated patterns and optimized by a genetic algorithm. Composite materials of GFRP and CFRP are incorporated to improve the durability of the proposed absorber. From the full-wave simulation, more than 90% absorption rate bandwidth is computed from 2.2 to 18 GHz such that the fractional bandwidth is about 156% for the incidence angles from 0° to 30°. Absorptivity is measured using the Naval Research Laboratory (NRL) arch method in an EM anechoic environment. It was shown that the measured results correlated with the simulated results. In addition, the proposed absorber underwent high temperature and humidity tests under military environment test conditions in order to investigate its durability.

## 1. Introduction

Recently, radar absorbing materials (RAM) and radar absorbing structure (RAS) technologies are becoming an important research topic named as radio frequency (RF) stealth. Here, the RF stealth refers to a technology that absorbs electromagnetic waves to evade radar signals. To avoid detection by electromagnetic waves of the radar system, various electromagnetic (EM) wave absorbers based on advanced technologies are proposed. Also, as radar system technology advances, the performance requirements for the EM absorbers are increasing. In order to secure these requirements, research on EM absorbers using various conductors, multilayer structures, and metasurfaces is being actively studied.

Several research studies on reduction of a radar cross section (RCS) have been proposed with metasurfaces based on pairs of anti-phase unit-cells. The pairs of unit-cells are designed to have the opposed reflection phase in target frequency ranges. By diffusion of the random array super-cells, the scattered RCS can be reduced by more than 10 dB over the frequency range from 13 to 32 GHz [1], 5 to 34 GHz [2], and 3.5 to 16.6 GHz [3]. However, in these cases, the RCS at limited angles of incidence can be suppressed, so that, it is difficult for these absorbers to reduce the averaged RCS over the three dimensionally spherical coordinates.

With respect to the RCS level of antennas, Danpeng Xie et al. suggest a multi-resonant gridded-square design as an absorbing ground. The measured maximum RCS reduction of the antenna with the proposed surface is 5 dB at 2–2.7 GHz [4]. H. B. Baskey et al. showed that a hybrid absorber was effectively employed for the RCS reduction of wide band Vivaldi antenna. The designed antenna achieved 10 dB RCS reduction in a frequency range of 6–12 GHz [5]. Juan Yang et al. presented a design of thin broadband absorbers using a periodic array of double square loops operating at 8–18 GHz. The loops are printed on a dielectric and four lumped resistors are added to four sides of the loop [6]. Amit Kumar Singh et al. described a dual-band EM absorber for C- and X-band applications which consists of two circular ring resonators with a thickness of λ0/142.65 at the lowest frequency. Its absorption is larger than 90% at 4.19 and 9.75 GHz with bandwidth of 665 MHz and 860 MHz, respectively [7]. These proposed absorbers utilize a conventional metallic pattern with an etching process and a surface mount device technology with lumped elements such as capacitor, resistor, diode, etc. Although this design can be realized with low cost, it has limited performance in terms of bandwidth extension.

Focus on the design of realistic perspective and feasibility, a conventional microwave absorption based on printed circuit board (PCB) was performed in the C-, X-band region [8,9,10]. A single layered low-profile broadband co-polarization absorber was proposed and achieved more than 90% of absorption over the whole band from 5.6 to 9.1 GHz [8]. A meta-dome structure was presented for ultra-wideband (4.6–12 GHz) radar absorption using an FR-4 protective layer with lumped resistors [9]. To reduce the RCS level, a snowflake shaped unit-cell was proposed on the dielectric substrate. The absorptivity of 98% was achieved at 10.52 GHz with the normal incidence case [10]. Several research studies in designing an electromagnetic absorber using printing technologies with a resistive ink or paste have been introduced [11,12,13,14,15,16,17,18]. Electromagnetic absorbers were presented using a transparent conductor with relatively high resistive losses [11,12,13,14]. An absorber using indium tin oxide (ITO) films was reported [11,12]. The proposed absorber achieved a 90% absorption bandwidth within 2.53–8.94 GHz and high angular stability (60°) [11]. Qian Zhou et al. proposed a sandwich absorber that could realize more than 90% absorption over 8–18 GHz for both TE- and TM-polarization when the incident angle was less than 30° [12]. Absorbers based on graphene were developed for high transparency and good flexibility in practical applications in the microwave band [13,14]. This was a broadband study involving controlling the sheet resistance of the graphene.

To overcome the narrow bandwidth characteristic, a carbon paste for the printed conductive patterns of EM absorber was used [15,16,17,18]. Inherently inhomogeneous structure based on 4-split ring resonators as the unit-cell was proposed and with the structure working in the 6.9–29 GHz range more than 80% absorption rate could be easily fabricated through conventional inkjet printing processes [15]. Y. Kim et al. presented a metasurface pattern designed from the genetic algorithm; the 90% absorption bandwidth was confirmed in the 8.8–11.6 GHz range, for which the fractional bandwidth was 27.5% for both TE- and TM-polarization with incident angles from 0° to 60° [16]. Y. Kim et al. proposed an ultra-wideband metamaterial absorber that consisted of double layer metasurfaces optimally designed by the genetic algorithm and printed using carbon paste. Based on the simulation results it was shown that, the 90% absorption bandwidth was obtained from 6.3 to 30.1 GHz of which the fractional bandwidth was 130% for the normal incidence [17]. The absorber covered the 10.4–19.7 GHz range with effective absorption (above 90%) and it was printed on flexible substrate using graphene nano-flake conductive ink through a stencil printing method [18].

Even though the performance of EM absorbers using resistive ink and paste have been verified successfully, there is a limitation in applying to a realistic structure due to mechanical problems such as non-flexibility, weight of the structure, environmental resistance, etc.

To surmount the limitations of single-layer and planar structures, multi-layer and three-dimensional absorbers were introduced [19,20,21,22,23,24,25]. A multi-layer and 3D structure-base with a switchable absorber was proposed because of diode bias voltage, but the structure could not avoid becoming a bigger size [19,23]. Houdi Xiao et al. introduce a broadband absorber with a 3-layer stacked absorber with ITO films [20]. The measured bandwidth of the absorption was more than 85% from 6.0 to 16.7 GHz. Yajuan Han et al. presented a reduced RCS patch antenna via dispersion engineering. The proposed structure resembled a pyramidal structure absorber [21]. Yuxuan Ding et al. proposed an ultra-wideband frequency-selective absorber designed by an adjustable and highly selective notch with multi-stacked layers. As a result, the absorption frequencies achieve about 4 to 18 GHz [22].

A double-sided parallel-strip line (DSPSL) was proposed by Guo Qing Luo et al. employed to produce the required infinite impedance of the absorber [24]. The proposed absorber operated at 1.5–12.3 GHz. Won-Ho Choi et al. designed a 3D honeycomb structure with glass/epoxy-MWCNT prepreg through an autoclave process. The absorption of the proposed honeycomb absorber was 3–16 GHz for more than 90% [25].

However, there are drawbacks in that the total size of the structure is bulky and/or the fabrication process becomes complex as the number of stacked layers is increased. To solve these problems, studies on EM absorbers in the form of composites to improve practicality have also been published [26,27,28,29,30]. 

In this paper, an ultra-wideband absorber is proposed with metasurfaces and a composite material for high practicality. The proposed absorber consists of double layer metasurfaces that are accomplished using commercial carbon paste with a screen-printing technology. The metasurfaces are composed of conductive pixelated patterns. The pixelated patterns were designed and optimized. To validate performance of the proposed absorber, the full-wave simulation based on the finite element method (FEM) was used. The bandwidth of 90% absorptivity was verified from 2.2 to 18 GHz, that is the fractional bandwidth is about 156% for both the transverse electric (TE) and magnetic (TM) polarizations (full-wave simulation only). The absorber was measured for incident angles of 0°, 7°, 15°, 30°, and 45° with TE polarization and it was found that the simulated and measured results agreed very well with each other. Also, the proposed absorber has good environmental resistance in the form of composite material and strong mechanical performance for various applications. Therefore, the proposed composite material absorber could be a good candidate to make a breakthrough in the RF stealth area to decrease the radar cross-section (RCS) of surveillance and reconnaissance platforms as well as the electromagnetic interference problems of wireless communication, satellite communication, etc.

## 2. Design of the Ultra-Wide Band Absorber

In this section, the layer configuration of the proposed EM absorber is introduced. Among various approaches for designing and optimizing the absorber, most of the variables were designed and optimized through optimization techniques based on several intuitive approaches of humans for the proposed ultra-wide band absorber. Therefore, the proposed genetic algorithm (GA) and the optimized two metasurface patterns through GA need to be explained. 

### 2.1. Layer Configuration of EM Absorber

An ultra-wideband EM absorber has a configuration of composite layers as shown in Figure 1. The absorber consists of the two thin metasurfaces of lossy conductive pixels, a GFRP layer, a CFRP layer, and several dielectric substrate layers. Design procedures of the pixelated metasurfaces are discussed in the next section. One of the metasurfaces, the lower layer, is inserted between the first and second dielectric layers. The other metasurface, an upper metasurface, is placed between the second and third dielectric layers. The first to third dielectric layers are made from a polymethacrylimide (PMI) form. The fourth dielectric layer, the GFRP layer, is added on the third dielectric layer to protect all structures from external forces. The thickness of the first and second dielectric layers is 5 mm, the third layer 2 mm, and the fourth layer 1 mm, respectively. The bottom of the overall structure is a high conductive CFRP layer serving as a reflector. The total thickness of the structure on the reflector was chosen to be 13 mm on excluding the protective fourth and third layers, the actual thickness is 10 mm up to the upper metasurface, which is equivalent to 0.06 λ_0_ at the lowest target frequency, 2 GHz, where λ_0_ is the wavelength in a free space. Since the dielectric constant of the form is similar to that of the free space, the proposed absorber has physically a very thin thickness. The complex relative permittivity of the PMI form was assumed to be 1.107 − j0.016, and the CFRP layer 4.6 − j0.253.

From now on, the reason for the layer configuration in Figure 1 of the proposed absorber is discussed as follows. First, the proposed design was based on durability and environmental resistance, therefore the absorber includes robust composite materials. The multi-layered absorber of PMI form, GFRP, and CFRP underwent an autoclave process at 125 °C. Second, a practical aspect was considered to fabricate the composite materials such as the protective layer, the form substrate, adhesive materials, etc. in order to be directly applied to stealth warships and aircrafts. All materials were purchased from well-known suppliers. Third, a multi-layered structure was proposed to achieve broadband characteristics. The principle of the operating mechanism in the ultra-wideband is as follows and described in Figure 1a. An incident wave on the top of the composite absorber interferes with the wave transmitting through the pattern of the bottom layer after being reflected backwards by the reflective layer. Because the phase shift of the wave is π (180°) when it is reflected from a reflector, destructive interference can occur when there exists a phase difference of π between the incident and reflected waves. While the EM waves are reflected from the metasurfaces or are transmitted through them, the phases of waves can be changed abruptly by interacting with the metasurfaces. As a result, the phase condition for destructive interference can be achieved by designing the patterns of the metasurfaces optimally under the configuration of the total structure. Based on the proposed composite absorbers’ multi-reflection as a backbone concept, the proposed absorber works and satisfies the absorption performance in the target frequency band as one composite structure.

### 2.2. Proposed Genetic Algorithm

This section describes an optimization algorithm, the so-called genetic algorithm. To design the optimal metasurfaces shown in Figure 2, a genetic algorithm that can find the ultimately optimized metasurface patterns mimicking the meiosis of the chromosome was proposed.

Figure 2 shows the flow chart of the genetic algorithm to design the pixelated metasurfaces. To achieve the design goals, a condition for evaluating performance of metasurfaces is required and there is a step to determine whether the goal has been achieved or not. In the beginning of the algorithm, two random 1-bit quantized chromosomes need to be generated. A dominant factor is selected from the two chromosomes, and then the next generations are prepared by crossover and mutation operations from the chromosomes. In the next step, the performance of the chromosome is evaluated repeatedly to find the optimal dominant gene. The cost function for the evaluation is as follows:$$\mathrm{Cos}t\;\mathrm{func}\mathrm{tion}\;=\frac{1}{N}\sum_{n=1}^{N}W_{1}\left|\Gamma_{f1}\right|+W_{2}\left|\Gamma_{f2}\right|+\cdots W_{n}\left|\Gamma_{\mathrm{fn}}\right|$$
where, Γ_fn_ is a reflection coefficient of the selected nth frequency and W_n_ is a weighing factor for the nth reflection coefficient. The factor of the evaluation is an arithmetic mean of reflection coefficients in the frequency range from 2 to 18 GHz. Reflection coefficients of the absorber are computed from a full-wave electromagnetic simulation, COMSOL Multiphysics [31]. Finally, with the optimal gene, the validation of the finalized results is completed. So, an optimized result can be obtained through the above procedure.

Figure 3 shows a schematic of an example of the proposed pixelated pattern for the metasurfaces. As mentioned above, the upper and lower metasurface layers are discretized into 20 × 20 square pixels of size 1 × 1 mm^2^. The gap between the neighboring unit-cells is 1 mm at the external boundaries of the square pixel arrays. The reason for the size and number of pixels is that the target lowest frequency band of the proposed absorber is 2 GHz, so the half-wavelength at 2 GHz is about 7.5 cm. Since the goal is to reduce the area and thickness of the unit cell to less than lambda/5, the unit cell was set to 20 × 20 mm^2^, and the pixel was set to 1 mm in consideration of the appropriate number of variables within the unit cell size. 

To optimize the 20 × 20 pixels, in the first step, 55-random 1-bit arrays are generated. Each bit is filled with ‘zero’ or ‘one’ corresponding to the area of dielectric or conductive ink. An example is shown in the Figure 2 inset. The final pattern is built by performing *x*-axis symmetry and then *y*-axis symmetry with respect to the pattern of the first quadrant. A random bit array is created iteratively and evaluated for optimization by the genetic algorithm. Therefore, to find the global optimum, 110-binary variables are needed for the upper and lower layers while at the same time for the absorber structure an iterative computation is performed. As a result, an optimized combination of metasurface satisfying the criterion of the figure of merit, in this case less than −10 dB reflectance, in the target frequency range, is obtained.

## 3. EM Simulation Results

To optimize and verify the proposed design method, a full-wave simulation is carried out using COMSOL Multiphysics. As for the full-wave simulation set-up and analysis, a general periodic boundary condition (PBC) is used for the pixelated unit-cell based on a finite element method (FEM). To simulate the proposed structure with PBC, the plane wave incident port was set to the same size as the area of the unit cell. Also, to reduce the simulation time, the bottom side of the absorber was considered as the PEC. It was assumed that there are no transmitted electromagnetic waves through the perfect electric conductor (PEC) on the bottom of the absorber. As a result, the reflectance of the absorber is computed by the follow equation, A = 1 − R − T, where ‘A’ is the absorption level, ‘R’ means the absolute value of reflection coefficient, and ‘T’ presents the absolute value of the transmission coefficient.

The optimized pattern of the proposed metasurfaces is shown in Figure 4a. To verify the proposed metasurfaces and absorber design, 3D full-wave simulation with a numerical analysis is proposed. Figure 4a,b show the full-wave simulated reflectance results. 

Figure 4b shows the results of the reflectance obtained with a change in sheet resistance from 40 to 50 ohm/sq with an increment of 5 ohm/sq for a normal incidence. Here, the incident wave is TE polarized, and the electric and magnetic fields are perpendicular to the plane of incidence of y- and x-axes, respectively. The simulation results for all three cases show a reflectance less than −10 dB, corresponding to absorption of more than 90%, in the target frequency band from 2.2 to 18 GHz. Therefore, it can be expected that this design is robust against variation of sheet resistance due to manufacturing errors of the metasurfaces.

Next, an absorption ratio for the sheet resistance of 40 ohm/sq was studied further with changes of polarization and incident angles. Figure 5a,b show the EM simulated reflectance dependent on the incident angle (θ) equal to 0°, 7°, 15°, 30°, and 45° in the case of the TE and TM polarization, respectively. As shown in Figure 5a, in all cases except for the case of the 45° oblique incidence, the reflectance is lower than −10 dB at the frequencies of 2.2–18 GHz. The results of reflectance for the TM polarization are observed in Figure 5b. Due to the symmetrical structure, the same results are obtained for the normal incidence between the TE and TM polarizations, while the absorption of 90% or more is satisfied from 0° to 45°.

Figure 6 presents the magnitudes of the electric-field (E-field) and magnetic-field (H-field) distribution with an xz-plane as the x-and y-component, respectively. The incident polarization is the same as the coordinate system in each figure. The plotted E-field is divided into three sections along with the *z*-axis. The first section corresponds to the area over the upper metasurface as a protective layer, in which electromagnetic waves over the air are incident. The second section is the area between the upper and lower metasurfaces as a dielectric, and the last section is the area between the lower metasurface and the reflector. For the lower frequency bands case as Figure 6a, the destructive interference is mainly created by the upper metasurface. This is because the path difference of the multi-reflected EM waves between the reflection due to the upper metasurface and the reflection caused by the floor reflector at 180 degrees with a certain height or more must be secured. So, the E-and H-field distribution in Figure 6a tells us that the overall thickness of the proposed composite absorber has sufficient thickness up to the lowest frequency of about 2 GHz. On the contrary, for the higher frequency bands, the lower metasurface can influence the relatively short wavelength. As a result, Figure 6c,d show more active interference at the media of the second and last one.

## 4. Fabrication and Measurement

To verify the characteristics of the designed absorber with a measurement, the metasurfaces are printed on a polyimide (PI) film and then added into a composite structure to produce test samples as shown in Figure 7a,b. The printing process is accomplished with a commercial carbon paste in the well-known silk-screen printing technology. The thickness of the PI film is 50 μm and the sheet resistance of the upper and lower metasurfaces is 40 ohm/sq.

To validate the durability under harsh conditions, environmental tests were conducted with the proposed electromagnetic composite absorber based on the metasurfaces. For these environmental tests, the temperature and humidity tests were performed using the facility as shown in Figure 8a. As for the high- and low-temperature tests, the methods and procedures are based on Method 501.7, Procedure I according to the MIL-STD-810H. Also, the humidity test followed Method 507.6, Procedure II from the MIL-STD-810H. The detailed specifications of the performed experiment are shown in Table 1.

Figure 8b,c show the absorption rate before and after the environmental tests of the proposed absorber as a sub-array structure where the size of sample is 15 × 15 cm^2^. The procedure of the tests was as follows (Initial of the under-test sample measurement → Temperature test → Measurement → Humidity test → Measurement). The environmental tests were completed through the above process, and the measured results showed no significant difference in terms of performance before and after the tests. However, due to the small-sized test samples, the measured reflection coefficient results have an uncertainty because of the 15 × 15 mm^2^ size below 6 GHz range. This is because the sample was not enough to satisfy the area for the NRL method. So, the main focusing region of the reflectance results are above the 6 GHz region. As a result, the measured reflectance results tell us that the proposed composite absorber has good robustness for temperature and humidity variation equivalent to MIL-STD-810H.

After conducting the environmental tests, to verify the lower band performance of the proposed EM absorber, the fabricated unit absorbers were arranged and measured in an anechoic chamber as shown in Figure 9.

An NRL arch measurement system was used to measure the reflectance of the absorber, as shown in Figure 9. The measurement system consists of two wideband horn antennas (transmitter and receiver) as the bistatic RCS on an arch fixture and a test sample holder on the ground. The position of the horn antennas determines the incident or reflective angles of the electromagnetic waves with respect to the sample holder. The minimum incident angle is 7 degrees. The metallic plate for calibration and a test sample are placed and tested on the sample holder. The physical dimension of the sample holder is 60 by 60 in square centimeters. The measurement system can measure 1–40 GHz, and the standard complies with IEEE STD 149 [32].

The reflectance responses of the fabricated absorber with respect to four incident angles (7°, 15°, 30°, and 45°) with the TE polarization were measured on an NRL Arch measurement system, and the results are shown in Figure 9. The measurement frequency range was from 2 to 18 GHz, and the polarization TE. As shown in Figure 10a, absorption of more than 90% can be seen in the target band from 2.2 to 18 GHz except for the 45° incidence angle. Even in the case of the 45° incidence angle, excellent performance is secured in certain frequency bands (from 6 to 16 GHz). Figure 10b shows the comparison between the simulation and the measurement results with respect to the incident angles and showing good correlations. Through these results, it was shown that the proposed method is valid for the design of ultra-wideband absorbers while the absorption performance was verified. Finally, an environmental test was conducted to see if the proposed composite-based absorber could perform a role as a composite material. The proposed absorber has distinct features as illustrated in Table 2 below.

## 5. Conclusions

An ultra-wideband electromagnetic multi-layered composite absorber with resistive conductor linked with two metasurfaces is proposed. The proposed absorber has good absorption property at the target frequency band with quadruple absorption points of resonance. To meet the ultimate goal, a genetic algorithm was used as the design method for the two metasurfaces. Through the proposed optimization algorithm, pixelated pattern generation and EM analysis are iteratively performed until an optimized result is obtained with the proposed cost function. As a result, ultra-wideband absorption performance is achieved as an integrated composite material. In addition, for TE-polarization, through the proposed mechanism, a bandwidth that satisfies 90% absorption can be achieved from 2.2 to 18 GHz of which the fractional bandwidth is 156% for not only the normal but also the oblique incidence angle (0° to 30°). To verify the proposed absorber design and performance, the absorptivity was measured by the NRL arch method with an anechoic environment. Through the measurement and full-EM simulation, the proposed absorber was confirmed, and comparable results were obtained. Furthermore, from this study, the proposed composite absorber of this design method has an extremely high applicability to actual structures such as tanks, airplanes, ships, etc. It was confirmed through environmental tests that the proposed absorber is reasonable for various applications. In addition, the proposed silk-screen printing method provides the lowest complexity of the fabrication process among methodologies adopted to fabricate multilayer structures. This increases the possibility of mass and large area production. Therefore, the proposed design method may break through the conventional concept that the number of absorption resonances coincides with that of the layers. Finally, based on the proposed metasurfaces, the based composite absorber achieves a down-to-earth breakthrough in realizing a low-profile, lightweight, durable, large-area expandable, and low-cost EM wideband absorber in the RF stealth field.

## Figures and Tables

**Figure 1 materials-16-05916-f001:**
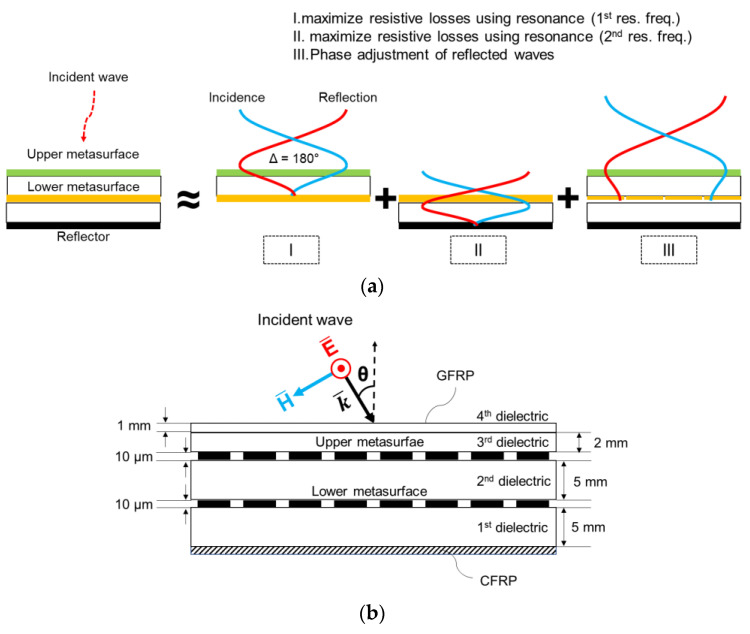
Proposed composite absorber. (**a**) Principle of operating mechanism, (**b**) Layer configuration.

**Figure 2 materials-16-05916-f002:**
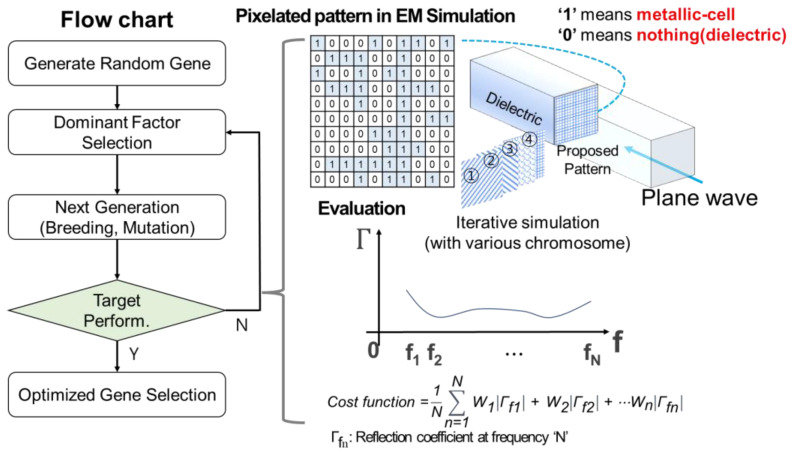
Flow chart of the proposed genetic algorithm.

**Figure 3 materials-16-05916-f003:**
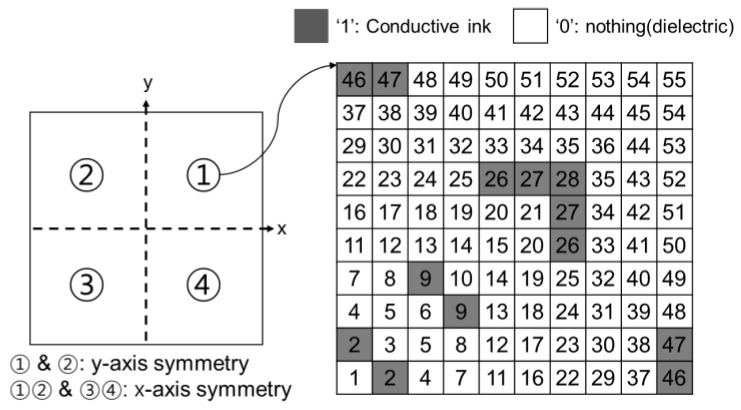
Schematic of the proposed pixelated pattern.

**Figure 4 materials-16-05916-f004:**
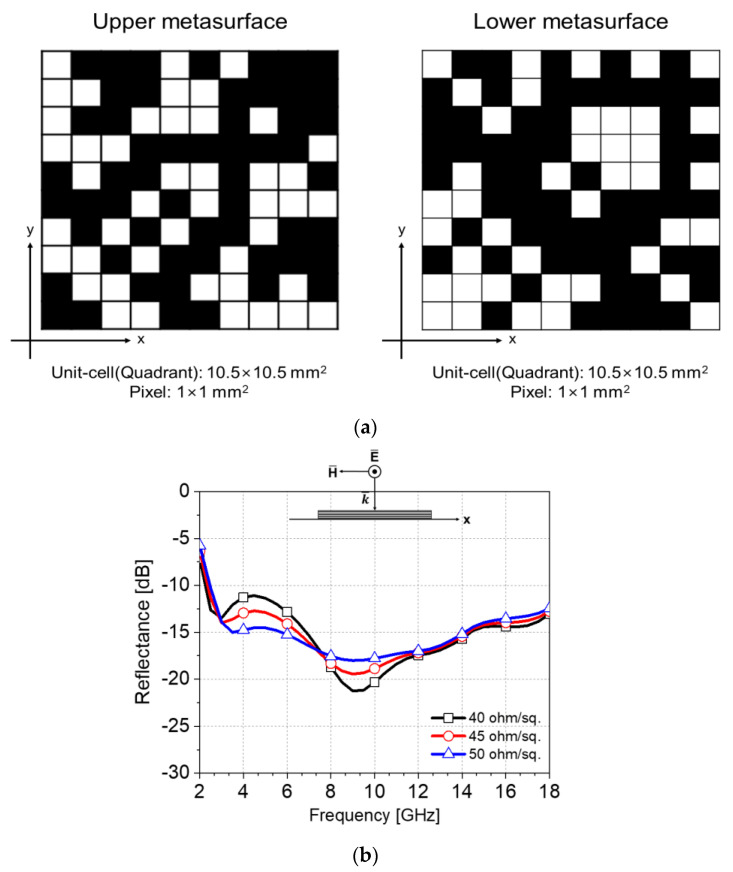
Schematics for optimized metasurfaces and EM simulation results of optimized metasurface absorber. (**a**) Left—upper metasurface pattern, right—lower metasurface pattern, (**b**) Simulated reflectance with respect to the sheet resistance at theta = 0 deg.

**Figure 5 materials-16-05916-f005:**
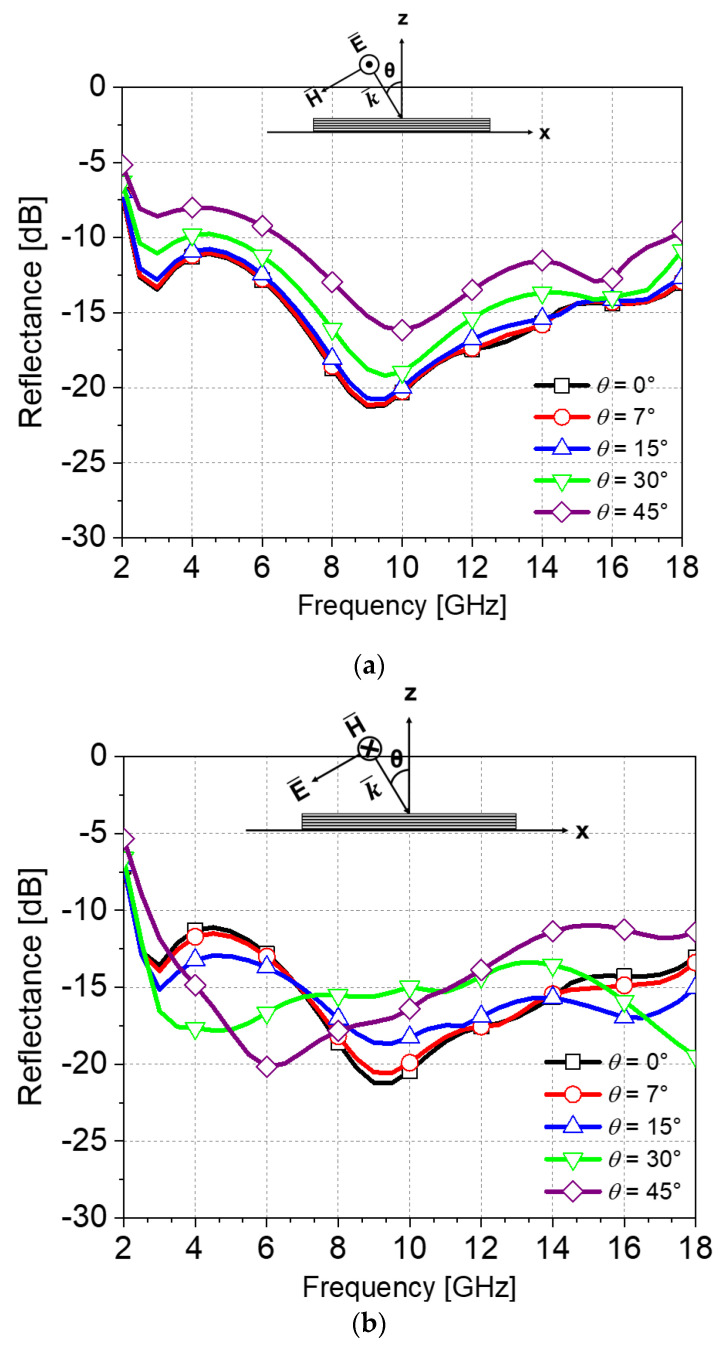
Simulated reflectance according to various incident angles. (**a**) TE-polarization, (**b**) TM-polarization.

**Figure 6 materials-16-05916-f006:**
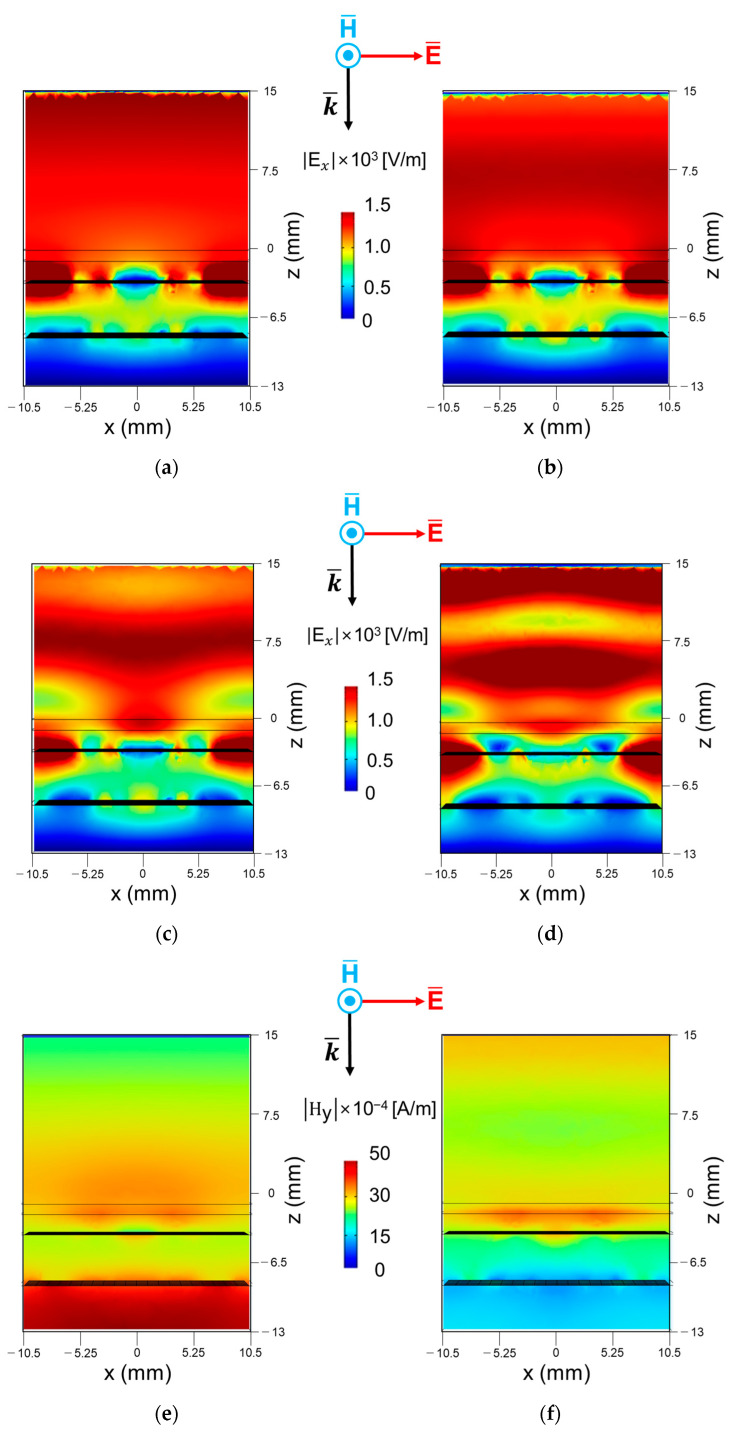
Magnitudes of the electric-field distribution along the x-and *z*-axis |Ex| at y = 0 mm at (**a**) f = 3 GHz, (**b**) f = 8 GHz, (**c**) f = 13 GHz, (**d**) f = 18 GHz, |Hy| at y = 0 mm at (**e**) f = 3 GHz, (**f**) f = 8 GHz, (**g**) f = 13 GHz, (**h**) f = 18 GHz.

**Figure 7 materials-16-05916-f007:**
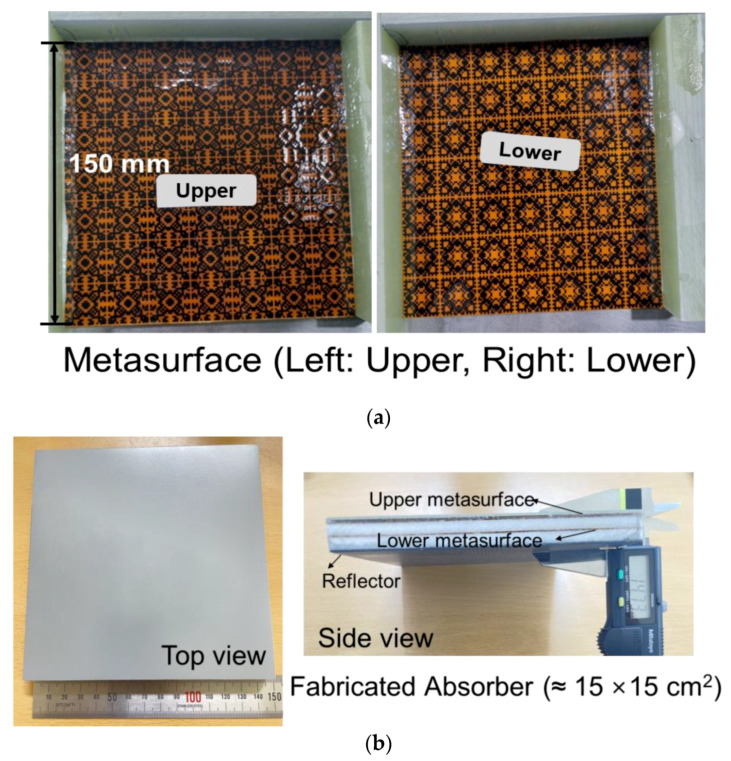
(**a**) Metasurfaces—top and bottom pattern, (**b**) Composite absorber.

**Figure 8 materials-16-05916-f008:**
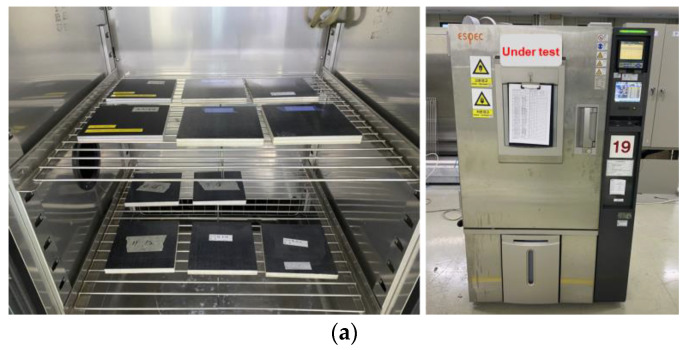
Environment test (**a**) Set-up, (**b**) Temperature test results (solid: before, dotted: after), (**c**) humidity test results (solid: before, dotted: after).

**Figure 9 materials-16-05916-f009:**
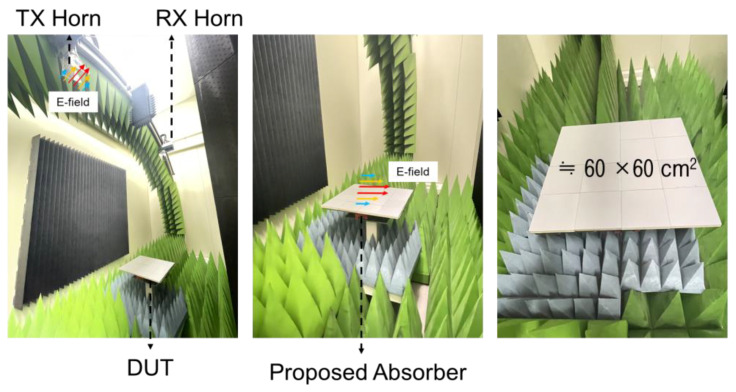
NRL arch measurement system with the proposed absorber.

**Figure 10 materials-16-05916-f010:**
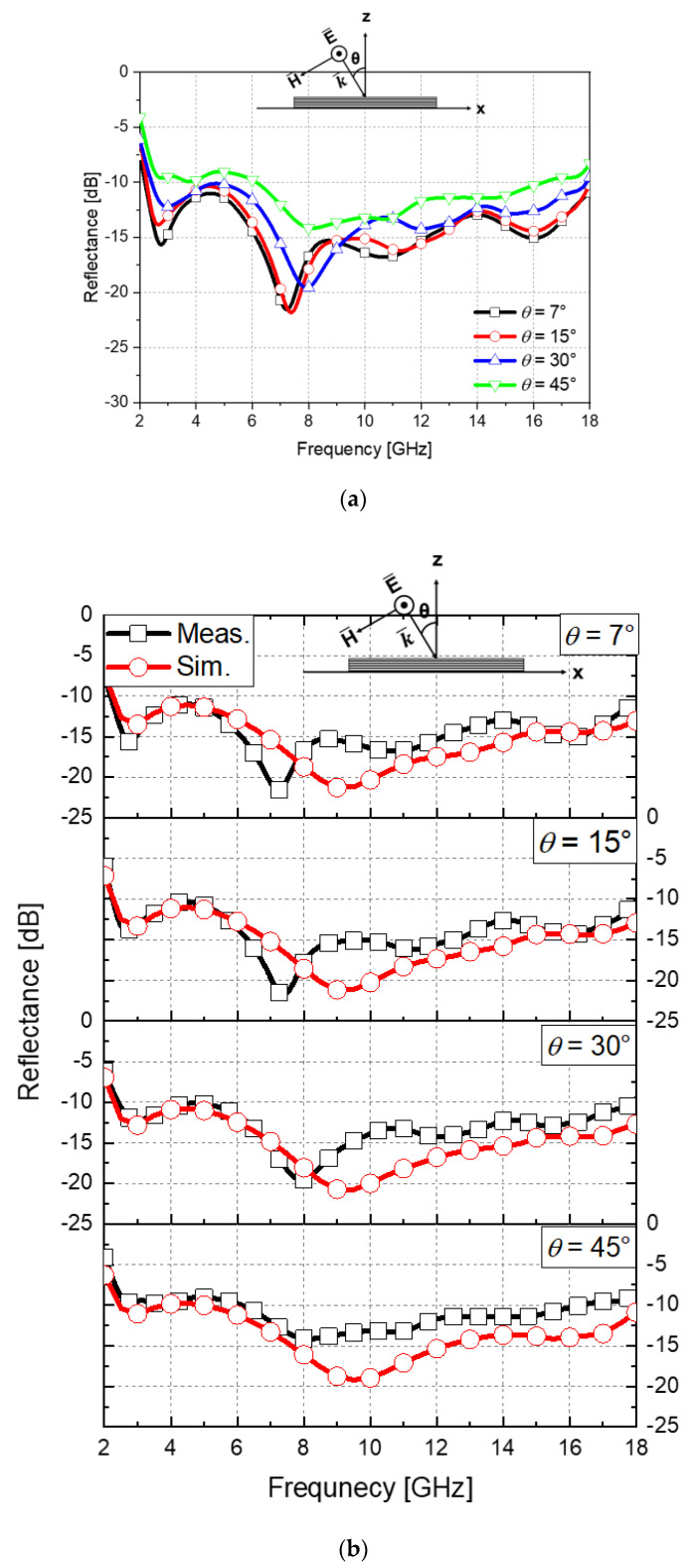
Measured reflectance results. (**a**) According to the incident angles, (**b**) Compared with the simulated results by COMSOL Multiphysics.

**Table 1 materials-16-05916-t001:** The conditions of the environment tests.

	Temperature Test	Humidity Test
Temperature range	30–63 °C (<30 °C/min)	30–60 °C
Humidity range	5–44%	95% *
Iteration number	7	10
Period	24 h	24 h

* Maintain the relative humidity at 95 ± 4 percent at all times except that during the descending temperature

**Table 2 materials-16-05916-t002:** Comparison of the proposed composite absorber and previous works.

Ref.	Feasibility	Lowest Freq. (FBW)	Size *	Materials
[33]	×	3.9 GHz (101%)	0.3 × 0.3 × 0.12 λ^3^	Carbon-loaded ABS
[34]	× ^#^	1.7 GHz (101%)	0.06 × 0.06 × 0.08 λ^3^	FR4/Chip R, L **/PIN diode
[35]	△	1.85 GHz ^†^ (105%)	0.06 × 0.06 × 0.14 λ^3^	Form & FR4/Chip R
[36]	△	1.89 GHz (113%)	0.25 × 0.25 × 0.1 λ^3^	Form & FR4/Chip R
[37]	×	0.21 GHz (130%)	0.02 × 0.02 × 0.005 λ^3^	NiZn Ferrite
[38]	△	2 GHz (160%)	0.13 × 0.13 × 0.13 λ^3^	Carbon black
[39]	△	5.8 GHz (71%)	0.5 × 0.5 × 0.1 λ^3^	PET, PDMS/CNT
[40]	×	1 GHz (93%)	0.078 × 0.078 × 0.067 λ^3^	FR4, Spacer
This work	○	2.2 GHz (156%)	0.15 × 0.15 × 0.09 λ^3^	Composite/Carbon paste

×: Low (Lumped elements or bias are required, no planar structure). △: Mid (No planar structure, additional protective layers are required). ○: High (possible to external exposure, low density, used actual reflectors). *: Unit-cell size based on the lowest freq. wavelength at the operating freq. **: Lumped resistor and inductor. ^†^: Absorbance of over 85%. ^#^: A multi-layer structure, and a spacer is required due to the PIN diode.

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
