# Peer review of "Ultra-Wideband Electromagnetic Composite Absorber Based on Pixelated Metasurface with Optimization Algorithm"

_materials, 2023, doi:10.3390/ma16175916_

Round 1

Reviewer 1 Report

In this paper, the author proposed an ultra-wideband electromagnetic absorber with more than 90% absorption rate bandwidth from 2.2 to 18 GHz. The fractional bandwidth is about 158% for the incidence anlges from 0to 30 degrees. Although the manuscript has been well written, there are some questions needed to be addressed.

1. How are the thicknesses of each dielectric layer and metasurface selected? Detailed explanation is needed.

2. Why did the author use a pixel size of 1 mm and a total number of 55 pixels?

3. In Fig. 7a, why do the patterns of the pixel of the upper metasurface and the lower metasurface look different. The circuit design of the patterns of the metasurfaces should be discussed.

4. As for the TM polarization, how is the measurement compared to the simulation? 

5. There exist some typos and incorrect sentences, which must be improved.

There exist some typos and incorrect sentences, which must be improved. Moderate editing of English language is required. 

Author Response

Thank you for your time and service.

Reviewer 2 Report

In this manuscript, authors present an ultra-wideband microwave absorber with a composite of two thin metasurfaces, 4 dielectric layers, a glass fiber reinforced polymer (GFRP), and a carbon fiber reinforced polymer (CFRP) which works as a conductive reflector. The thin metasurfaces are accomplished with the 1-bit pixelated patterns and optimized by a genetic algorithm. The composite materials of GFRP and CFRP are incorporated to improve durability of the proposed absorber. From the full-wave simulation, more than 90 % absorption rate bandwidth is computed from 2.2 to 18 GHz such that the fractional bandwidth is about 158 % for the incidence angles from 0° to 30°. The subject of the paper is interesting. However, here are some problems need to be addressed.

1. The English writing should be improved, there are some errors.

2. It should be given some comment about how to design the proposed structure.

3. More details about simulation should be given, for example, what kind of boundary conditions, what kind of wave source. 

4. As shown in Figs.6, authors just present the electric-field distribution along with the x-and z-axis, what about magnetic-field distribution?

5. How to get the sheet resistance of the optimized metasurfaces in experiment? What is the practical value of the shee resistance?

6. There are some discrepancies between simulation and measurement, resonable explanations are necessary.

7.What is the advantages of the proposed absorber compared to the previous design?

It is better to compare the performance at the end of the paper to reflect the research value.

8. The authors should reference other absorber published recently.

In this manuscript, authors present an ultra-wideband microwave absorber with a composite of two thin metasurfaces, 4 dielectric layers, a glass fiber reinforced polymer (GFRP), and a carbon fiber reinforced polymer (CFRP) which works as a conductive reflector. The thin metasurfaces are accomplished with the 1-bit pixelated patterns and optimized by a genetic algorithm. The composite materials of GFRP and CFRP are incorporated to improve durability of the proposed absorber. From the full-wave simulation, more than 90 % absorption rate bandwidth is computed from 2.2 to 18 GHz such that the fractional bandwidth is about 158 % for the incidence angles from 0° to 30°. The subject of the paper is interesting. However, here are some problems need to be addressed.

1. The English writing should be improved, there are some errors.

2. It should be given some comment about how to design the proposed structure.

3. More details about simulation should be given, for example, what kind of boundary conditions, what kind of wave source. 

4. As shown in Figs.6, authors just present the electric-field distribution along with the x-and z-axis, what about magnetic-field distribution?

5. How to get the sheet resistance of the optimized metasurfaces in experiment? What is the practical value of the shee resistance?

6. There are some discrepancies between simulation and measurement, resonable explanations are necessary.

7.What is the advantages of the proposed absorber compared to the previous design?

It is better to compare the performance at the end of the paper to reflect the research value.

8. The authors should reference other absorber published recently.

Author Response

Thank you for your time and service.

Round 2

Reviewer 2 Report

Accept

Accept

Author Response

thank you